# Awareness, Attitudes and Clinical Practices Regarding Human Papillomavirus Vaccination among General Practitioners and Pediatricians in Switzerland

**DOI:** 10.3390/vaccines9040332

**Published:** 2021-04-01

**Authors:** Levy Jäger, Oliver Senn, Thomas Rosemann, Andreas Plate

**Affiliations:** Institute of Primary Care, University Hospital Zurich, 8091 Zurich, Switzerland; oliver.senn@usz.ch (O.S.); thomas.rosemann@usz.ch (T.R.); andreas.plate@usz.ch (A.P.)

**Keywords:** human papillomavirus, vaccine, primary care providers, survey

## Abstract

In Switzerland, the human papillomavirus vaccination (HPVv) coverage rate lies below a desirable threshold. General practitioners (GPs) and pediatricians have been recognized as important providers of the HPVv, but there is little known about their self-attributed role and its relationship with their actual HPVv behavior. Therefore, the objective of this study was to explore the awareness, attitudes, and clinical practices of Swiss GPs and pediatricians concerning HPVv by means of a web-based questionnaire. We analyzed the responses of 422 physicians (72% GPs, 28% pediatricians). A substantial proportion of respondents considered the HPVv “absolutely essential” (54.2% of pediatricians, 30.6% of GPs). GPs indicated spending more time and effort on HPVv counseling for female rather than male patients more often compared to pediatricians (44.0% versus 13.9%, *p* < 0.001). The weekly number of patients aged 18–26 years seen in practice (*p* = 0.002) and whether the HPVv was deemed “absolutely essential” (adjusted odds ratio 2.39, 95% confidence interval 1.12–5.08) were factors associated with GPs administering HPVv in their practice. Shortcomings in terms of awareness, effort in the identification of potential vaccination candidates, and the role of male patients were revealed. By addressing these gaps, Swiss primary care providers could contribute to an increase in the national HPVv coverage rate.

## 1. Introduction

Infection with human papillomavirus (HPV) has long been identified as a major cause of anogenital cancer, and HPV-related diseases are associated with high mortality and morbidity worldwide [1,2]. Various studies have demonstrated the role of the HPV vaccine in the prevention of precancerous lesions and invasive cervix cancer [3,4,5]. One of the World Health Organization global targets for 2030 in the fight against sexually transmitted infections (STIs) is a 90% HPV vaccination (HPVv) coverage [6]. In the United States and Europe, HPVv coverage rates vary greatly and seldom exceed an 80% margin [7,8]. In Switzerland, HPVv has been included in the vaccination schedule since 2007 [9]. Administration of the HPV vaccine is currently recommended as a primary vaccination for girls aged 11–14 years, as a catch-up vaccination for girls aged 15–19 years, and as a supplementary vaccination for women aged 20–26 years and males aged 11–26 years [10]. HPVv coverage has remained comparatively low despite its introduction over one decade ago, with 59% of girls aged 16 years and 17% of boys aged 16 years having received two vaccine doses as of 2019 [11].

Primary care providers like general practitioners (GPs) and pediatricians have been recognized as important prescribers of the HPVv. In Switzerland, pediatricians generally provide primary care only to children and adolescents up to the age of 18 years, while just a minority of GPs address both this age group and adults [12]. A national survey in 2014 showed that GPs and pediatricians were responsible for 20% and 12%, respectively, of HPV vaccine prescriptions among 11–19-year-old girls, while gynecologists and school physicians administered the greatest proportions of HPV vaccines in this population group (32% and 28%, respectively) [13]. These results revealed that the role of primary care providers for HPVv could be enhanced, especially regarding catch-up vaccinations by GPs for young women and men who missed HPV immunization by school physicians or pediatricians earlier in life.

Previous studies have assessed primary care providers’ knowledge, awareness, attitudes, and clinical practices in the context of HPVv, revealing, among others, the factors associated with HPV recommendation are behavior, knowledge, and opinions about HPV vaccine utility [14,15,16]. However, assessments of primary care providers’ self-attributed role as HPVv providers and its relationship with their actual HPVv behavior are lacking. Therefore, the objective of our study was to explore attitudes of Swiss primary care providers about the HPVv and clinical practices of its prescription and counseling behavior by means of a cross-sectional survey. In addition, we aimed to explore awareness of HPV-induced disease and HPVv indications, the perceived suitability of general practice as a setting for HPV vaccine administration, and any perceived patient barriers to the HPVv.

## 2. Methods

### 2.1. Population and Instrument

We implemented the survey as a self-administered web-based questionnaire using the SurveyMonkey web tool (SurveyMonkey Inc., San Mateo, CA, USA). Questions were developed after a review of literature and data concerning HPVv practices in Switzerland [13,17,18,19]. Before study initiation, 13 GPs, pediatricians, and general practice residents affiliated with our institution piloted the questionnaire and tested it for comprehensibility, duration of completion, and content validity. Their feedback was implemented in the final version of the questionnaire.

Invitations to our questionnaire were sent by the Association of Swiss General Practitioners and Pediatricians (Hausärzte Schweiz—mfe) as part of their regular electronic newsletter followed by mail invitations to all practice addresses contained in their mailing list. As an incentive, respondents were given the opportunity to take part in a lottery (one of five tablet computers). We provided both a web link and a quick response code for access to either of the two language versions (German and French). No reminders were sent, and the survey was closed 17 weeks after the distribution of invitations. Users with the same internet protocol address were prevented from answering the survey more than once in order to avoid duplicates. A checklist for reporting results of internet e-surveys (CHERRIES, according to [20]) is provided in Appendix A. Respondents were asked for consent on the opening page of the survey, where information about the study and its aims were presented. By selection of a tick box, they provided consent for use and analysis of their responses as described. All data were analyzed anonymously.

The survey comprised of three sections, each addressing a different aspect of the HPVv. Section 1 collected data concerning participants’ registration and attitudes towards cantonal HPVv programs (not included in this analysis due to its specificity for the Swiss healthcare system, results are shown in Appendix A). Section 2 addressed the awareness and knowledge of HPV-induced disease and current recommendations concerning the HPVv in Switzerland. One item in this section asked for primary care providers’ attributed relevance to 12 different vaccinations, including the HPVv, of the Swiss vaccination schedule. To evaluate the attitudes with regard to HPV vaccine prescription, respondents were asked to indicate the clinical settings they considered to pose the greatest opportunity to reach eligible patients belonging to four different population groups. Section 3 concerned clinical practices of HPVv counseling and prescription. Finally, the survey collected demographic and professional characteristics of respondents. Survey questions and responses included in our analysis are summarized in Table 1 and Table 2 and Figure 1, Figure 2 and Figure 3 of the Results section.

### 2.2. Statistical Analysis

The descriptive statistics are presented in terms of proportions for categorical variables, while medians and corresponding interquartile ranges (IQR) were computed for continuous variables. If not indicated otherwise, stratification into primary care settings (general practice and pediatrics) was carried out for descriptive analyses. Open-ended text responses were extracted and categorized inductively into a flat coding frame [21]. Categories encompassing at least 5% of overall responses were further considered for analysis.

To assess the associations and differences among primary care settings, a dichotomous variable encoding perceived the relevance of the HPVv (deemed HPVv “absolutely essential” on a 6-point Likert scale) was modeled by means of multiple logistic regression on respondents’ primary care setting (general practice, pediatrics) with an adjustment for gender and experience as a primary care provider in years. We further analyzed the relationship between GPs’ status as HPVv providers (offers versus does not offer HPVv) and their self-attributed relevance as HPVv providers for patients aged 11–17 and 18–26 years (defined as whether respondents indicated their own setting among the ones deemed appropriate for HPVv for the respective age group) by means of logistic regression. Gender, experience as a primary care provider in years, numbers of patients aged 11–17 years and 18–26 years seen in an average week, and HPVv deemed “absolutely essential” were used for further adjustment. No model used respondent age as a covariate given its high correlation with the experience as a primary care provider (Pearson’s correlation coefficient 0.85). The results of the regression models were reported in terms of odds ratios (OR) with corresponding 95% confidence intervals (CI). The significance of categorical model covariates with more than two levels was assessed by means of Wald chi-squared tests and corresponding *p*-values. Further differences in proportions and distributions among the two provider groups were assessed by means of Pearson’s chi-squared and Wilcoxon rank-sum tests, respectively.

The statistical significance was assessed by means of *p*-values with a threshold of 0.05. We exported response data as comma-separated files from the SurveyMonkey servers and analyzed them with R 4.0.3 (R Foundation for Statistical Computing, Vienna, Austria). The figures were edited with the packages ggplot2 and likert [22,23,24].

## 3. Results

### 3.1. Respondent Characteristics

In total, 463 respondents answered to the consent form on the first survey page, of which 422 (91%) filled in at least 75% of the remaining questionnaire and were included as study participants (Table 1). Of these, 304 (72.0%) were GPs and 118 (28.0%) were pediatricians. The overall median age was 55 years (IQR 48–60 years), 39.1% of included respondents were female, and the median experience as a primary care provider was 20 years (IQR 11–25 years). Of the study participants, 82.9% indicated offering HPV vaccines in their practice at all, pediatricians more often than GPs (*p* < 0.001).

### 3.2. Awareness and Knowledge of HPV-Associated Disease and HPVv Indication

Full awareness of HPV’s association with cervix cancer and genital warts was reported by 99.3% and 96.6% of respondents, respectively (Figure 1a). The correct fractions of 80–100% of all cervix cancer and genital warts cases attributable to infection with HPV types covered by the nine-valent vaccine (based on [5]) were indicated by 31.9% and 36.0% of respondents, respectively (Figure 1b). The indication of HPVv attributed to different age groups, taking into account the history of sexual intercourse, is shown in Figure 1c.

### 3.3. Attitudes in the Context of HPVv

The HPVv ranked fifth out of 12 vaccinations in terms of attributed importance with 97% of respondents considering it “absolutely essential,” “very important,” or “important” (Figure 2). Upon multivariable adjustment, the HPVv was more likely to be deemed “absolutely essential” by pediatricians than by GPs (OR 2.63, 95% CI 1.67–4.15, Table 3).

Figure 3 shows the healthcare settings deemed to pose the greatest opportunity to reach different age and gender groups (females and males aged 11–17 years and 18–26 years) for HPV vaccines. Overall, the GP was chosen most frequently for males aged 18–26 years (97.1% of respondents), while being chosen less often (73.0% of respondents) than the gynecologist (90.0% of respondents) for females aged 18–26 years. For both females and males aged 11–17 years, the pediatrician was indicated most frequently (82.3% and 81.4% of respondents, respectively), followed by the school physician (67.8% and 67.3% of respondents, respectively).

### 3.4. Clinical Practices of HPVv

The numbers of administered HPV vaccines in the 4 weeks prior to the questionnaire response as well as reported HPVv counseling behavior are presented in Table 2. Pediatricians indicated a higher number of HPV vaccines administered during the 4 weeks prior to the questionnaire response than GPs (median 10, IQR 5–18, and median 2, 1–4, respectively, *p* < 0.001). In addition, pediatricians reported higher estimated frequencies of counseling about HPV and associated disease (*p* < 0.001) and about its prevention by means of the HPVv (*p* = 0.014) to potential HPVv candidates. Frequencies of counseling about the importance of HPVv prior to first sexual intercourse did not differ significantly between GPs and pediatricians (*p* = 0.082). The reported reasons for refusal of a recommended HPVv, which was asked as an open-ended text question, could be classified into six categories. In particular, the most commonly indicated reasons were lack of information and insight concerning the benefits of the HPVv (41.3% of respondents), general skepticism towards vaccines (28.9% of respondents), and concerns regarding the safety of the HPV vaccine (26.7% of respondents).

The results of the multivariable logistic regression for HPVv providing among GPs are reported in Table 4. The weekly number of patients aged 18–26 years (*p* = 0.002) and deeming the HPVv “absolutely essential” (OR 2.39, 95% CI 1.12–5.08) were associated with higher chances of providing the HPVv. The association to deeming general practice appropriate for HPV vaccine administration was not significant.

## 4. Discussion

In Switzerland, primary care has been identified as a major health care setting that could increase national vaccination coverage rates [13]. Therefore, in this explorative study, we assessed the awareness, attitudes, and clinical practices of prescription and counseling behavior of Swiss GPs and pediatricians in the context of HPVv. We found that the HPVv was well-known and accepted, but the reported number of HPVv doses administered was low. Furthermore, we identified shortcomings in terms of knowledge, identification of potential candidates for vaccination, and the importance of male patients.

A recent systematic review has revealed a mismatch between clinicians’ support for the HPV vaccine in general and their vaccination recommendations and vaccination rates [25]. We can confirm this discrepancy, as both GPs and pediatricians acknowledged the importance of the HPVv, and both groups of primary care providers saw themselves as relevant providers for HPV vaccine administration. In addition, the self-reported estimated frequencies of counseling about HPV, HPV-related diseases, and the importance of the HPVv prior to first sexual intercourse were high. The low reported vaccination numbers were opposite to these numbers, especially by GPs. Thereby it seems that the basic requirement of meeting a relevant number of potential vaccination candidates in practice is fulfilled. In our study, GPs reported seeing a substantial number of adolescents and young adults in their daily practice, and the caseload was identified as a relevant determinant of HPVv in our regression analysis. Against the background of the low vaccination coverage rates, one can conclude that a relevant proportion of eligible patients are seen every day by GPs in Switzerland. Most GPs reported checking the vaccination status during admission of new patients and during check-up visits; however, 20% reported not doing so. Moreover, not even half of the respondents indicated travel medicine counselling as a situation in which HPVv status is usually checked. Counseling about HPV is part of general STI advice that has been recommended to be included in every travel medicine consultation [26]. Hence, check-up visits and travel medicine counselling seem to pose frequently unused opportunities to check HPVv status. Accordingly, HPVv counseling would profit from the integration of protocols for such routine consultations, including occupational health screening or military recruitment visits.

The reported awareness on HPV and associated diseases was high, but the estimated proportion of preventable disease cases by vaccination was clearly underestimated. Various studies have shown vaccination providers’ knowledge gaps [14,27,28], and insufficient knowledge on HPV and the HPVv has been identified as a personal barrier to providing the HPVv [25]. Furthermore, it became apparent among the survey respondents that patient age and gender strongly influence their attitudes. Half of the respondents deemed HPVv indication for patients aged 18–26 years with a history of sexual contacts to only be given on an individual basis or not any longer at all. Of course, the protective effect of the vaccination is higher if administered at a young age and before the first sexual contact, but protection against viral strains that there has been no contact with yet can still be given [3,29]. This is a reason why the HPVv is still recommended for these age groups [10]. Men are an important target group for the HPVv: It not only protects them from HPV-related cancer and genital warts, but the vaccination of men also contributes to herd protection in the community [30]. Our observation that only 50% of all GPs (79% of pediatricians) reported having spent time and effort on HPVv counseling for both genders equally provides further opportunities to increase HPVv coverage rates in the future. However, the focus of vaccination on girls and young women and younger rather than older patients observed in our study is a known phenomenon and has already been reported in the literature [25].

Lack of information, vaccine skepticism, and concerns about vaccine safety were the most commonly reported reasons for HPVv refusal. These findings are in line with known patient-related barriers preventing vaccination and highlight the need for comprehensive knowledge of primary care providers, as HPVv providers’ communication seems to be crucial to overcoming patient-related concerns [25].

Across most aspects of our study, pediatricians tended to report higher efforts in the HPVv and showed higher knowledge compared to GPs. The physician specialty is a well-known provider factor, and our results reflect the findings of other studies [16,31]. Their exact reasons remain unclear. One explanation could be that, as outlined in the Introduction section, the HPVv is categorized differently depending on age and sex in the Swiss vaccination schedule. Evidence supports the finding that the HPVv is often recommended weakly compared to other vaccines in the sense of an optional vaccine [25,32,33]. However, whether and to what degree these formal differences in the Swiss vaccination schedule could affect physicians’ attitudes needs to be evaluated in further analyses.

Our study has revealed important shortcomings in the context of HPVv in the primary care setting. We believe the most important finding is the discrepancy between primary care providers’ awareness and support for HPVv in general and their vaccination recommendations and rates. The identified shortcomings should be addressed in future interventions, with a focus on the following aspects: (i) knowledge on the role of the HPVv in the context of HPV infection and HPV-related diseases, (ii) improvement of the identification of unvaccinated patients, (iii) highlighting the importance of male patients. In this context, it is important to consider that knowledge about HPV and the effectiveness and safety of the HPVv is a major facilitator in the process of vaccination [25]. In addition, evidence has shown that both the approach and style of HPVv recommendations influence the vaccination’s success, suggesting communication skills as a target for potentially effective interventions [34,35].

### Strengths and Limitations

A major strength of our study is the comprehensive evaluation of different domains of high relevance for HPVv among health care providers. In particular, discrepancies between self-attributed roles and actual clinical practices in the context of HPVv have rarely been investigated in the literature. Furthermore, by addressing both GPs and pediatricians, our analysis provides an exhaustive comparison between two major groups of primary care providers involved in HPVv administration in Switzerland.

On the other hand, our study displays some limitations. First, we are unaware of the exact response rate as we were not granted access to the mailing list the Association of Swiss General Practitioners and Pediatricians used to send survey invitations. We can only provide a crude estimate for the total number of invitation recipients from the association’s annual report of 2019, where a total of 4501 members was listed, thus yielding an approximate response rate of 10.3% [36]. It must be noted that the mailing list included retired physicians and physicians not working in the primary care setting any longer. Furthermore, we could not determine the number of recipients who had taken notice of the invitations, which is why this value most likely underestimates the true response rate. However, comparing the demographic and professional characteristics of our respondents with those reported by a representative study assessing the workforce in Swiss primary care [37], we found a high accordance in terms of age, gender, and practice characteristics. Second, due to the limited response numbers, further subgroup analyses, such as the influence of the language region (German-speaking versus French-speaking parts of Switzerland), were not feasible. Third, we are aware that potential sources of bias might lie in the participants’ self-selection and the inaccuracy of self-reporting, which are issues typical of web-based surveys [38].

## 5. Conclusions

Our study highlights the opportunity of the primary care setting to contribute to an increase in national HPVv coverage rates. HPVv is well-known and accepted, but shortcomings in terms of knowledge, the identification of potential candidates for vaccination, and the importance of male patients could be identified. Based on the existing knowledge of the HPVv and HPV-associated diseases among Swiss primary care providers, the specific gaps we identified in this study should be increasingly incorporated into continuing medical education efforts.

## Figures and Tables

**Figure 1 vaccines-09-00332-f001:**
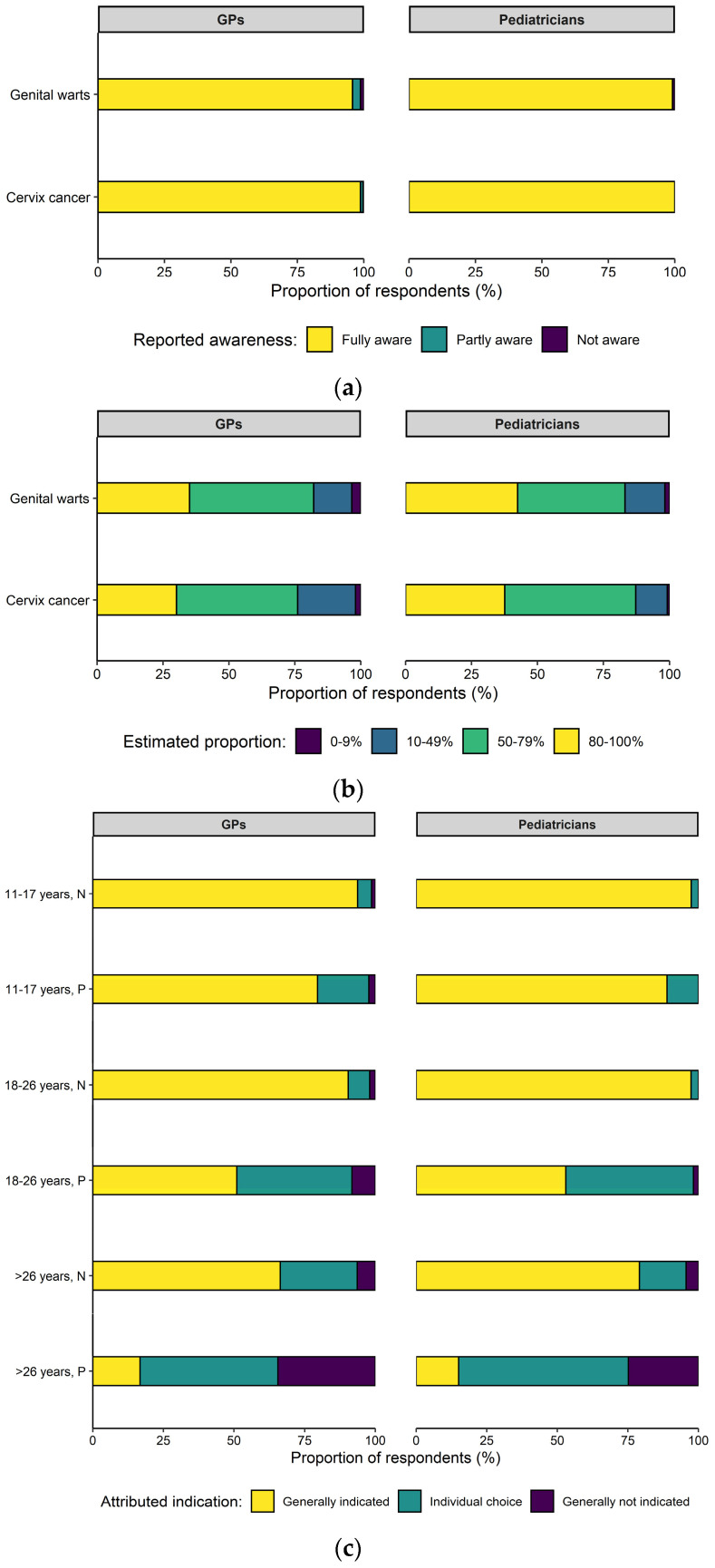
(**a**) The reported awareness of association of human papillomavirus (HPV) infection with cervix cancer and genital warts. (**b**) For the same diseases, the estimated fraction of cases preventable by the currently available nine-valent HPV vaccine. (**c**) The indication attributed to HPV vaccination for different population groups. Abbreviations: general practitioners (GPs); negative history of sexual intercourse (N); positive history of sexual intercourse (P).

**Figure 2 vaccines-09-00332-f002:**
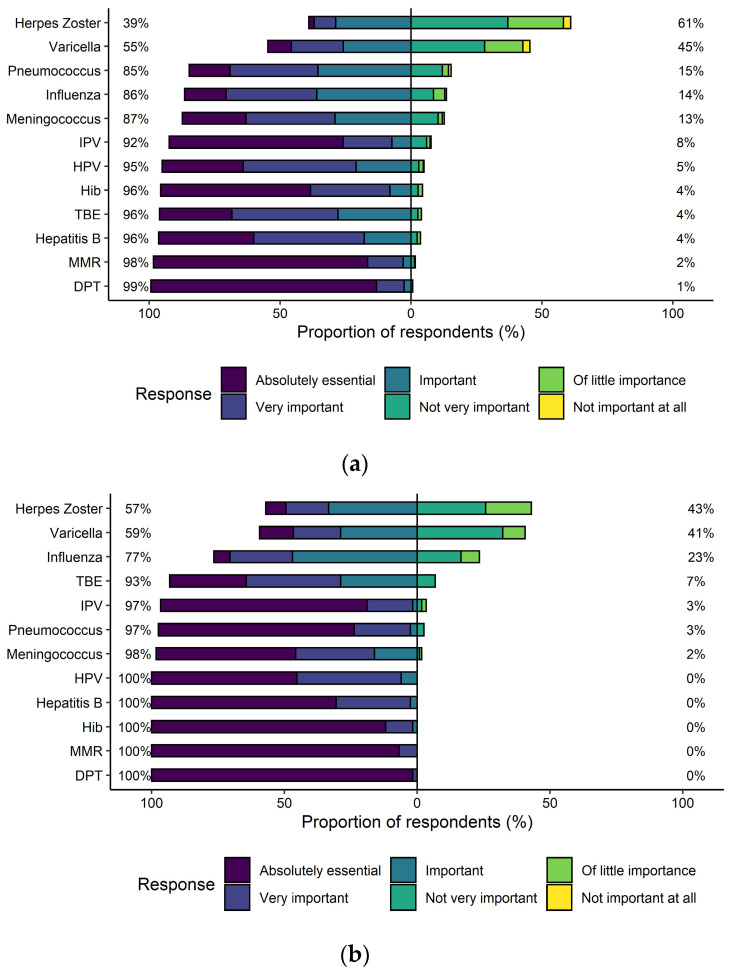
The relevance attributed to recommendations of the Swiss vaccination schedule [10] with the responses stratified into general practitioners (**a**) and pediatricians (**b**). Abbreviations: diphtheria, pertussis, tetanus (DPT); *Haemophilus influenzae* type b (Hib); human papillomavirus (HPV); inactivated polio vaccine (IPV); measles, mumps, rubella (MMR); tick-borne encephalitis (TBE).

**Figure 3 vaccines-09-00332-f003:**
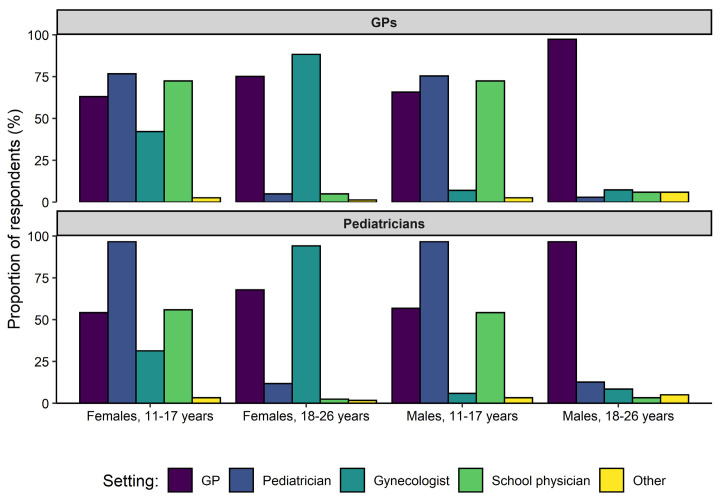
Evaluation of various clinical settings in the context of human papillomavirus (HPV) vaccine administration. For this item, respondents were asked to select the clinical settings they deemed to pose the greatest opportunity to reach four different age and gender groups for HPV vaccination (multiple choice). Abbreviations: general practitioner (GP).

**Table 1 vaccines-09-00332-t001:** Demographic and professional characteristics of included survey respondents (*n* = 422). Abbreviations: general practitioner (GP); interquartile range (IQR); human papillomavirus (HPV).

Characteristic	GPs *n* = 304 (72.0%)	Pediatricians *n* = 118 (28.0%)	Missing Data (% of Total)
**Age in years, median (IQR)**	55 (48–61)	52 (47–58)	6.2
Gender			2.1
Male, *n* (%)	190 (63.8)	55 (47.8)	
Female, *n* (%)	105 (35.2)	60 (52.2)	
Other, *n* (%)	3 (1.0)	0 (0.0)	
Years of experience as a primary care provider, median (IQR)	20 (11–26)	17.5 (12–24)	3.8
Type of practice, *n* (%)			1.9
Single	73 (24.5)	25 (21.6)	
Double	68 (22.8)	40 (34.5)	
Group	157 (52.7)	51 (44.0)	
Language region			0.0
German	256 (84.2)	95 (80.5)	
French	48 (15.8)	23 (19.5)	
Number of patients aged 11–17 years seen in typical week			0.0
≤5	185 (60.9)	13 (11.0)	
6–10	93 (30.6)	25 (21.2)	
11–20	22 (7.2)	56 (47.5)	
>20	4 (1.3)	24 (20.3)	
Number of patients aged 18–26 years seen in typical week		Not asked	0.0
≤5	57 (18.9)		
6–10	124 (41.1)		
11–20	82 (27.2)		
>20	39 (12.9)		
Part-time percentage in %, median (IQR)	80 (60–100)	75 (60–90)	1.4
Offers HPV vaccine in their practice at all	235 (77.3)	115 (97.5)	0.0

**Table 2 vaccines-09-00332-t002:** Questions directed at study participants who indicated providing human papillomavirus (HPV) vaccination in their practice (*n* = 350, 82.9% of all included respondents). Abbreviations: general practitioners (GPs); interquartile range (IQR).

Question	GPs *n* = 235 (78.6% of All)	Pediatricians *n* = 115 (97.5% of All)	Missing Data (% of Total)
Number of HPV vaccines administered in the past four weeks, median (IQR)	2 (1–4)	10 (5–18)	0.0
Estimated frequency of counseling about HPV and associated disease to potential vaccination candidates, *n* (%)			0.6
<10% of potential vaccination candidates	3 (1.3)	0 (0.0)	
10–39%	14 (6.0)	0 (0.0)	
40–60%	13 (5.6)	2 (1.8)	
61–90%	57 (24.4)	16 (14.0)	
>90%	147 (62.8)	96 (84.2)	
Estimated frequency of counseling about the prevention of associated diseases by HPV vaccination among potential vaccination candidates, *n* (%)			0.6
<10% of potential vaccination candidates	3 (1.3)	0 (0.0)	
10–39%	10 (4.3)	1 (0.9)	
40–60%	10 (4.3)	1 (0.9)	
61–90%	44 (18.8)	12 (10.5)	
>90%	167 (71.4)	100 (87.7)	
Estimated frequency of counseling about the importance of HPV vaccination prior to first sexual intercourse among potential vaccination candidates, *n* (%)			0.9
<10% of potential vaccination candidates	4 (1.7)	0 (0.0)	
10–39%	16 (6.9)	5 (4.4)	
40–60%	14 (6.0)	3 (2.6)	
61–90%	46 (19.7)	16 (14.0)	
>90%	153 (65.7)	90 (78.9)	
Gender group for which the most time and effort are spent on HPV vaccination counseling, *n* (%)			0.3
Both equally	118 (50.4)	91 (79.1)	
Females	103 (44.0)	16 (13.9)	
Males	13 (5.6)	8 (7.0)	
Neither	0 (0.0)	0 (0.0)	
Typical patient encounters in which HPV vaccination status is assessed (multiple choice), *n* (%)			0.0
Check-up visits	187 (79.6)	112 (97.4)	
Admission of new patients	186 (79.1)	103 (89.6)	
Travel medical advice	108 (46.0)	46 (40.0)	
Regular control visits	39 (16.6)	44 (38.3)	
Acute consultations	35 (14.9)	41 (35.7)	
Other	58 (24.7)	13 (11.3)	
I usually do not assess my patients’ HPV vaccination status	1 (0.4)	0 (0.0)	
Most common reasons for HPV vaccination refusal indicated by patients or their legal custodians (open-text item)			15.7
Lack of information or insight (especially concerning herd immunity, benefits for males)	82 (38.9)	48 (46.2)	
General skepticism towards vaccines	75 (35.5)	16 (15.4)	
Concerns regarding safety and adverse effects	44 (20.9)	40 (38.5)	
Moral concerns including fear of incitation of risk-taking sexual behavior	10 (4.7)	17 (16.3)	
Patients considered too young for discussion about HPV vaccination	5 (2.4)	18 (17.3)	
The number of recommended vaccinations is generally deemed too high	14 (6.6)	5 (4.8)	
Non-classifiable/other categories	21 (10)	2 (1.9)	

**Table 3 vaccines-09-00332-t003:** The multivariable logistic regression on study participants (*n* = 422) deeming HPVv “absolutely essential”. Abbreviations: confidence interval (CI); odds ratio (OR).

Variable	OR (95% CI)
Intercept	0.42 (0.25–0.70)
Setting: pediatrics (reference: general practice)	2.63 (1.67–4.15)
Experience as primary care provider in years	0.97 (0.62–1.52)
Gender: male (reference: female)	1.01 (0.98–1.03)

**Table 4 vaccines-09-00332-t004:** The multivariable logistic regression on human papillomavirus (HPV) vaccination offering for general practitioners (offers versus does not offer, *n* = 304). Abbreviations: confidence interval (CI); odds ratio (OR).

Variable	OR (95% CI)
Intercept	0.53 (0.07–4.04)
Gender: male (reference: female)	0.79 (0.39–1.60)
Experience as primary care provider in years	0.97 (0.94–1.00)
Number of patients aged 11–17 years seen in average week: >5 (reference: ≤5)	1.89 (0.80–4.46)
Number of patients aged 18–26 years seen in average week (reference: ≤5) *	
6–10	4.38 (1.66–11.55)
>10	3.52 (1.62–7.62)
Deems HPVv “Absolutely Essential”	2.39 (1.12–5.08)
Deems general practice appropriate for HPV vaccine administration to patients aged 11–17 years	1.67 (0.88–3.17)
Deems general practice appropriate for HPV vaccine administration to patients aged 18–26 years	2.36 (0.38–14.60)

* *p*-value (Wald chi-squared): 0.002.

## Data Availability

The data presented in this study are available on request and after approval of the internal review board from the corresponding author.

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
