# Peer review of "Awareness, Attitudes and Clinical Practices Regarding Human Papillomavirus Vaccination among General Practitioners and Pediatricians in Switzerland"

_vaccines, 2021, doi:10.3390/vaccines9040332_

Round 1

Reviewer 1 Report

I enjoyed reading about this interesting study in your paper and have some relatively minor suggestions about how the paper could be improved. Line and page numbers refer to those within the PDF.

Introduction section:

Pediatricians in your study were located in primary care services. Although this is how services are structured in Switzerland and is also common in the USA it is not universal. It is much less common in other European countries and the rest of the world where it is more usual for care to all age groups to be delivered (or coordinated) by a general practitioner or physician. As Vaccines is an international journal with an international readership it would be useful to comment on the Swiss context of pediatric primary care in this section.

Methods:

It is not clear if any changes were made to the questionnaire as result of feedback from the pilot. If so please include here or state that none were needed (line 63,page 2).

Results:

Some attention is need to the titles of all of the figures and tables and how they all appear in the text as this is confusing to read at the moment. All titles do not always clearly indicate what is presented (for example Table 2 does not indicate the data only refers to pediatricians) and the words "Error! Reference Source not found" appear everywhere one would expect to see the figure or table number suggesting formatting and style errors. Please review these and make sure they all have the correct reference and number within the main body of the text , and that all of their titles accurately reflect the content of the figures and tables.

Strengths and limitations:

This should include the limitations of (any) survey design that includes self- selection of participants and the accuracy of self-report.

General:

There are a few typos and unusual use of words that will need to be addressed at the final proof reading stage  (for example on lines 33-34, page 1; line 182, page 7; lines 186-187, page 8).

Author Response

Response to Reviewer 1 Comments

General point: I enjoyed reading about this interesting study in your paper and have some relatively minor suggestions about how the paper could be improved. Line and page numbers refer to those within the PDF.

General response: We thank the reviewer for appreciating our work and the constructive remarks, which we would like to address point-by-point.

Point 1, Introduction section: Pediatricians in your study were located in primary care services. Although this is how services are structured in Switzerland and is also common in the USA it is not universal. It is much less common in other European countries and the rest of the world where it is more usual for care to all age groups to be delivered (or coordinated) by a general practitioner or physician. As Vaccines is an international journal with an international readership it would be useful to comment on the Swiss context of pediatric primary care in this section.

Response 1: We agree that this aspect of primary healthcare is rather specific to Switzerland and might not be known by international readers. We added corresponding information to the Introduction section.

Page 1, lines 39-41: In Switzerland, pediatricians generally provide primary care only to children and adolescents up to the age of 18 years, while just a minority of GPs addresses both this age group and adults [11].

Point 2, Methods: It is not clear if any changes were made to the questionnaire as result of feedback from the pilot. If so please include here or state that none were needed (line 63,page 2).

Response 2:  We completed information about questionnaire piloting by making the following changes in the Methods section of the manuscript.

Page 2, lines 64-67: Before study initiation, 13 GPs, pediatricians and general practice residents affiliated with our institution piloted the questionnaire and tested it for comprehensibility, duration of completion and content validity. Their feedback was implemented in the final version of the questionnaire.

Point 3, Results: Some attention is need to the titles of all of the figures and tables and how they all appear in the text as this is confusing to read at the moment. All titles do not always clearly indicate what is presented (for example Table 2 does not indicate the data only refers to pediatricians) and the words "Error! Reference Source not found" appear everywhere one would expect to see the figure or table number suggesting formatting and style errors. Please review these and make sure they all have the correct reference and number within the main body of the text , and that all of their titles accurately reflect the content of the figures and tables.

Response 3: (See response for Reviewer 2, Point 4) We thank the reviewer for pointing out formatting and style errors that impair clarity. Unfortunately, we cannot trace back the source of these errors and believe that they occurred at some point during the submission process. The manuscript version we worked on for revisions had been cleaned of such errors by the editors. Furthermore, we would like to mention that Table 2 does not refer to pediatricians only, but to all survey respondents (one of the covariates included in the model is a binary variable indicating the primary care setting: pediatrics versus general practice). We changed the table caption to enhance comprehensibility as follows.

Page 7, lines 153-154: Multivariable logistic regression on study participants (n = 422) deeming HPVv “Absolutely essential”. Abbreviations: CI, confidence interval; OR, odds ratio.

Point 4, Strenghts and limitations: This should include the limitations of (any) survey design that includes self- selection of participants and the accuracy of self-report.

Response 4: We agree with the reviewer that the limitation due to self-selection bias reviewer is of major importance and that it can affect external validity of the study. We elaborated the issue by adding the following lines to this paragraph.

Page 11, lines 288-290: Third, we are aware that potential sources of bias might lie in self-selection of participants and inaccuracy of self-reporting, issues typical of web-based surveys [35].

Point 5, General: There are a few typos and unusual use of words that will need to be addressed at the final proof reading stage  (for example on lines 33-34, page 1; line 182, page 7; lines 186-187, page 8).

Response 5: We thank the reviewer for pointing out these flaws. Typos and lexical choices will be addressed at the final proof reading stage as suggested. At this point, we changed the sentences mentioned by the reviewer. In particular, we replaced the list of odds ratios for the levels of weekly number of patients aged 18-26 years by the overall Wald chi squared p-value for that covariate to increase readability (and added a corresponding sentence to the Methods section) and corrected a similar typo in the abstract of the article as follows.

Page 1, lines 17-20: The weekly number of patients aged 18–26 years seen in practice (p = 0.002) and whether HPVv was deemed “Absolutely essential” (adjusted odds ratio 2.39, 95% confidence interval 1.12–5.08) were factors associated with GPs administering HPVv in their practice.

Page 1, lines 35-37: Coverage of HPVv has remained comparatively low despite its introduction over one decade ago, with 59% of girls aged 16 years and 17% of boys aged 16 years having received two vaccine doses as of 2019 [10].

Page 3, lines 113-114: Significance of categorical model covariates with more than two levels was assessed by means of Wald chi squared tests and corresponding p-values.

Page 9, lines 190-193: The weekly number of patients aged 11–17 years (p = 0.002) and deeming HPVv “Absolutely essential” (OR 2.39, 95% CI 1.12–5.08) were associated with higher chances of providing HPVv.

Page 9, lines 193-196: The association to deeming general practice appropriate for HPV vaccine administration was not significant.

Reviewer 2 Report

The study is interesting and well described, here are some comments to improve its readability and understanding by the readership. My comments are directly included in the PDF text.

Author Response

Response to Reviewer 2 Comments

General point: The study is interesting and well described, here are some comments to improve its readability and understanding by the readership. My comments are directly included in the PDF text.

General response: We thank the reviewer for the encouragement and the constructive remarks, which we would like to address point-by-point:

Point 1, Abstract: I will add the most significant OR in abstract

Response 1: The reviewer wishes for presentation in the abstract of the most significant odds ratio as found in the regression models. In the abstract, we included the odds ratio for the significant effect we considered to be of greatest interest, namely the impact of deeming HPV vaccination “Absolutely essential” on general practitioners administering HPV vaccination, as follows (in addition, a typo concerning patient age was corrected).

Page 1, lines 17-20: The weekly number of patients aged 18–26 years seen in practice (p = 0.002) and whether HPVv was deemed “Absolutely essential” (adjusted odds ratio 2.39, 95% confidence interval 1.12–5.08) were factors associated with GPs administering HPVv in their practice.

Point 2, lines 61-63: this represents how many GPs and pediatricians who have tested the understanding of the questionnaire ?

Response 2: We provided this information in the Methods section as follows.

Page 2, lines 64-67: Before study initiation, 13 GPs, pediatricians and general practice residents affiliated with our institution piloted the questionnaire and tested it for comprehensibility, duration of completion and content validity. Their feedback was implemented in the final version of the questionnaire.

Point 3, lines 66-69: why the questionnaire was not translated into Italian, which is also a national language

Response 3: The reviewer remarks that a version of the questionnaire could have been implemented in Italian. We would like to mention that the newsletter that included advertisements for the survey as well as the mail invitation sent by the Association of Swiss General Practitioners and Pediatricians (Hausärzte Schweiz – mfe) were issued in German and French only, which put a constraint on our choice of language. In addition, French and German are, as of 2021, the only official course languages of undergraduate (Bachelor level) clinical medicine programs in Switzerland. Most recipients of survey invitations in Italian-speaking Switzerland were therefore be expected to be familiar with at least one of these national languages.

Point 4, lines 120-121: this error must be corrected

Response 4: (See response for Reviewer 1, Point 3) We thank the reviewer for pointing out formatting and style errors that impair clarity. Unfortunately, we cannot trace back the source of these errors and believe that they occurred at some point during the submission process. The manuscript version we worked on for revisions had been cleaned of such errors by the editors.

Point 5, Table 1: I am not sure that this missing data column is absolutely essential for reading of  this table

Response 5: We agree with the reviewer that the missing data column is not essential for the tables. However, we strongly believe that information about missing data contributes substantially to assessment of sources of bias that might affect interpretation of survey results. A high proportion of missing values for a single item can indicate, for instance, that the corresponding question might have been hard to understand and was therefore skipped by respondents. We believe that the missing data column does not relevantly impair legibility of the table and that it contributes to transparency of reporting.

Point 6, Figure 1: I find this figure not very readable, I think it should be modified to make it more readable. it should be separated in two with a and b on one side and c alone

Response 6: We thank the reviewer for pointing out poor readability in this figure. We separated the figure into a longer format as suggested, but will leave the final choice about formatting to the editors.

Point 7, Table 2: ORs presented in this table the crude ORs or adjusted ORs for significant variables in the univariate model?

Response 7: We deliberately restrained from a model selection approach based on significance in bivariable analyses, as it has repeatedly been discouraged by statistical experts [1,2]. However, we are well aware that this procedure is still in use in many areas of medical research. Given the small number of available covariates, we also restrained from covariate selection using a relaxed level of significance in bivariable analyses, as it would not add substantial power to our models and bears the potential of omitting variables relevant for model interpretation. As stated in the Methods section, the reported odds ratios were derived from the multivariable logistic regression models and are, as such, inherently “adjusted” to the model covariates [3].

Point 8, Table 2: if we have the CI at 95% we do not need the p for the interpretation of the significance

Response 8: We agree with the reviewer that duality of confidence intervals and p-values makes reporting both quantities redundant when it comes to mere assessment of significance at a given level. We removed the corresponding column, since the reported p-values were not discussed in the main text.

Point 9, Figure 3: I find this figure very unclear and understandable

Response 9: We thank the reviewer for pointing out lack of comprehensibility in this figure. We changed formatting in order to increase font size and legibility and changed the figure caption for clarification as follows.

Page 8, lines 166-169: Evaluation of various clinical settings in the context of human papillomavirus (HPV) vaccine administration. For this item, respondents were asked to select the clinical settings they deemed to pose the greatest opportunity to reach four different age and gender groups for HPV vaccination (multiple choice).

Point 10, Table 3: same comment as for table 2 concerning the missing column

Response 10: See our response to Point 5.

Point 11, Table 4: same comments as for table 3 concerning the OR and the p values

Response 11: See our response to Point 8.

Point 12, lines 266-281: there are two other limitations:

the Italian speaking GPs were not questioned.

a selection bias probably exists, because the GPs who answer this kind of online questionnaire on vaccinations are often the most convinced by vaccination or on the contrary the most reluctant to vaccination, so it is difficult to generalize these results to all GPs and pediatricians

Response 12: We thank the reviewer for pointing out two further limitations to interpretation of our results. Concerning the first limitation, that Italian-speaking physicians were not addressed directly, we would like to refer to our response to Point 2. While it is true that an Italian version of the questionnaire was not implemented, we believe that physicians in the Italian-speaking region of Switzerland who are members of the Association of Swiss General Practitioners and Pediatricians are familiar with at least one of German or French. Concerning the second limitation, we agree that selection bias is a common issue of web-based surveys and that it is very likely to have occurred in our study. We added the following lines to the Strengths and Limitations section of the Manuscript.

Page 11, lines 288-290: Third, we are aware that potential sources of bias might lie in self-selection of participants and inaccuracy of self-reporting, issues typical of web-based surveys [35].

References

  1. Heinze, G.; Dunkler, D. Five myths about variable selection. Transplant international : official journal of the European Society for Organ Transplantation 2017, 30, 6-10.
  2. Sun, G.W.; Shook, T.L.; Kay, G.L. Inappropriate use of bivariable analysis to screen risk factors for use in multivariable analysis. Journal of clinical epidemiology 1996, 49, 907-916.
  3. Stoltzfus, J.C. Logistic regression: a brief primer. Academic emergency medicine : official journal of the Society for Academic Emergency Medicine 2011, 18, 1099-1104.
  4. Kantonales Durchimpfungsmonitoring Schweiz. Available online: https://www.bag.admin.ch/bag/de/home/gesund-leben/gesundheitsfoerderung-und-praevention/impfungen-prophylaxe/informationen-fachleute-gesundheitspersonal/durchimpfung.html (accessed on 04 December 2020)
  5. Cartier, T.; Senn, N.; J, C.; Y, B. Switzerland; Kringos, D., Boerma, W., Hutchinson, A., Eds.; European Observatory on Health Systems and Policies: Copenhagen, Denmark, 2015; Available online: https://www.ncbi.nlm.nih.gov/books/NBK459012/ (accessed on 23 March 2021).
  6. Ball, H.L. Conducting Online Surveys. Journal of human lactation : official journal of International Lactation Consultant Association 2019, 35, 413-417.